# Enhancing Biocide Efficacy: Targeting Extracellular DNA for Marine Biofilm Disruption

**DOI:** 10.3390/microorganisms10061227

**Published:** 2022-06-15

**Authors:** Benjamin Tuck, Elizabeth Watkin, Anthony Somers, Maria Forsyth, Laura L. Machuca

**Affiliations:** 1Curtin Corrosion Centre, WA School of Mines: Minerals, Energy and Chemical Engineering, Curtin University, Kent Street, Bentley, WA 6102, Australia; benjamin.tuck@postgrad.curtin.edu.au; 2Curtin Medical School, Curtin University, Kent Street, Bentley, WA 6102, Australia; e.watkin@curtin.edu.au; 3Institute for Frontier Materials, Deakin University, Geelong, VIC 3217, Australia; anthony.somers@deakin.edu.au (A.S.); maria.forsyth@deakin.edu.au (M.F.)

**Keywords:** extracellular DNA, microbiologically influenced corrosion, biofilm, extracellular polymeric substances, EPS, corrosion inhibitor, biocide enhancement

## Abstract

Biofilm formation is a global health, safety and economic concern. The extracellular composition of deleterious multispecies biofilms remains uncanvassed, leading to an absence of targeted biofilm mitigation strategies. Besides economic incentives, drive also exists from industry and research to develop and apply environmentally sustainable chemical treatments (biocides); especially in engineered systems associated with the marine environment. Recently, extracellular DNA (eDNA) was implicated as a critical structural polymer in marine biofilms. Additionally, an environmentally sustainable, multi-functional biocide was also introduced to manage corrosion and biofilm formation. To anticipate biofilm tolerance acquisition to chemical treatments and reduce biocide application quantities, the present research investigated eDNA as a target for biofilm dispersal and potential enhancement of biocide function. Results indicate that mature biofilm viability can be reduced by two-fold using reduced concentrations of the biocide alone (1 mM instead of the recommended 10 mM). Importantly, through the incorporation of an eDNA degradation stage, biocide function could be enhanced by a further ~90% (one further log reduction in viability). Biofilm architecture analysis post-treatment revealed that endonuclease targeting of the matrix allowed greater biocide penetration, leading to the observed viability reduction. Biofilm matrix eDNA is a promising target for biofilm dispersal and antimicrobial enhancement in clinical and engineered systems.

## 1. Introduction

Microorganisms generate globally significant health and economic impacts due to biofilm formation. Biofilms can be described as complex microbial communities living at a solid/liquid interface embedded in protective polymeric substances. By forming on steels, biofilms invariably result in microbiologically influenced corrosion (MIC) or biofouling, especially in marine environments. Marine biofilms alone contribute between 20% and 40% to all corrosion prevention and maintenance costs [1,2], which is estimated at around US $4T per annum [3]. In most natural environments the biofilm is the preferred living arrangement of microorganisms, offering up to 1000 times greater chemical tolerance to bacterial cells [4,5]. Chemical tolerance in these populations is ultimately the result of species diversity and extracellular polymeric substances (EPS), especially in the marine environment. The EPS is a self-produced matrix comprised mainly of polysaccharides, extracellular DNA (eDNA) and proteins [6,7,8]. The molecular composition of the matrix is critical to the structure and integrity, as well as function, of biofilms.

Although EPS is important for cell adhesion/cohesion, horizontal gene transfer, metabolism and interspecies interactions, the composition of the EPS in marine environments has not been well described. Conversely, the composition of single and multi-species biofilms in clinical settings has been extensively studied over the past two decades, leading to notable breakthroughs in fundamental understanding and treatment of clinically significant biofilms. For example, using a DNA degrading enzyme, it was demonstrated that dispersal of *Pseudomonas aeruginosa* biofilms could be achieved [9]. Since this sensitivity to eDNA dispersal was first revealed, a number of publications have further demonstrated the structural role of eDNA in biofilms from clinical strains; including *Staphylococcus* sp. [10], *Enterococcus* sp. [11], *Pseudomonas* sp. [12], *Burkholderia* sp. [13] and others [14]. Based on the importance of eDNA in most clinically relevant biofilms, it was hypothesized that multi-species marine biofilms developed on metals would also rely on the structural stability afforded by eDNA. 

In the previous investigation of this chapter, multi-species biofilms were developed over six weeks in marine simulating conditions and evaluated for EPS composition. In agreement with the hypothesis and previous reports, eDNA was more abundant in the matrix than other macromolecules frequently identified in the EPS. The eDNA pool was also relatively stable over six weeks, indicating a long-term involvement in biofilm architecture. In the present study, marine biofilm eDNA was targeted for degradation to enhance the efficacy of a novel biocide compound.

Biocides are the primary defense against biofilms in marine environments. Engineered systems such as pipelines are susceptible to biofouling and MIC, and can be difficult to access for mechanical scrubbing (pigging) and inspection. Equipment and infrastructure must therefore rely on chemical dosing to ensure material longevity and functionality. Current biocide compounds are toxic and environmentally hazardous, and thus face growing global scrutiny [15]. However, replacement of biocides with contemporary environmentally sensitive options is a challenging task. Novel greener compounds must also be effective and economically viable. Of developing alternatives, multi-functional organic inhibitor compounds are a promising strategy [16,17,18]. With multiple functions (for example, corrosion inhibition and biocidal capacity), the number of compounds applied, and thus dosing costs, can be reduced. Recently, a 10 mM concentration of a novel, green corrosion inhibitor exhibited 96% corrosion inhibition efficiency on CS (AISI 1030) after 30 min of exposure [19]. The compound, CTA-4OHcinn, contains a hexadecyl trimethylammonium cation and a trans-4-hydroxy-cinnamate anion [19]. Quaternary ammonium salts are broadly applied as antimicrobial surfactants [19,20,21,22], with low toxicity when combined with the *trans*-4-hydroxy-cinnamate anion. Indeed, recent research demonstrates that the CTA-4OHcinn toxicity profile is reduced compared to cetrimonium bromide (CetBr), a common and safe antimicrobial additive in cosmetic products [23]. Established as an effective corrosion inhibitor and biocide [16,19,24], CTA-4OHcinn is a promising candidate for real-world applications.

The history of chemical treatment application has demonstrated that biofilms develop tolerance to biocides [25]. Glutaraldehyde (GLUT) is applied on a global scale to mitigate MIC, yet it faces efficacy challenges as a result of microbial tolerance acquisition. *Escherichia coli* for example utilizes overexpression of aldehyde reductases to impede GLUT activity [26]. Reports of single species biofilms capable of tolerating harsh chemical treatments are concerning, especially as natural biofilms contain complex communities that are known to develop tolerance at a faster rate. Thus, the present study anticipates tolerance acquisition to chemical treatments, with the aim of enhancing CTA-4OHcinn efficacy by targeting the EPS matrix.

Based on previous research identifying eDNA as a critical structural polymer in marine multispecies biofilms, the present research aimed to develop a method to evaluate enhancement of CTA-4OHcinn function. Mature, multispecies marine biofilms were developed and compared against biofilms treated with a low dose of CTA-4OHcinn. Subsequently, the same treatment was applied with an additional eDNA degradation stage. The results reveal, for the first time in marine conditions, that targeted biofilm dispersal approaches can enhance multifunctional biocide efficacy.

## 2. Materials and Methods

### 2.1. Microorganisms

A consortium comprising *Shewanella chilikensis* strain DC57, *Pseudomonas balearica* strain EC28 and a laboratory strain of *Klebsiella pneumoniae* was employed in this research based on growth characteristics described in previous work [25]. *S. chilikensis* DC57 and *P. balearica* EC28 were recently recovered from corroded steel in a marine industrial facility where MIC was implicated [26]. Pure bacterial strains were cultivated in artificial seawater (ASW) as previously described [27,28], and supplemented with Bacto™ casamino acids (10 mM, Thermo Fisher Scientific, Waltham, MA, USA), sodium pyruvate (10 mM), D  (+)  glucose (10 mM) and ammonium nitrate (NH_4_NO_3_30 mM). After 24–48 h of incubation at 30 °C, the cultures were harvested in log phase and manually counted using a Neubauer haemocytometer before washing twice in ASW at 12,000 rpm. Cell pellets were resuspended in ASW before inoculation into reactor media. The final cell number used for all reactors was 1 × 10^5^ cells/mL of each isolate.

### 2.2. Sample Preparation and Surface Finish

CS coupons (AISI 1030, AZo Materials, Manchester, UK) were prepared as previously described. Briefly, coupons were cut to produce a working surface of 1.34 cm^2^ and electrocoated using Powercron^®^ 600CX solution (PPG Industries, Pittsburgh, PA, USA). The working surface was then wet-ground using 120 g SiC sandpaper, rinsed in 100% ethanol and dried under nitrogen gas. Before and after fixing to reactor rods, coupons were irradiated with ultraviolet light (UV) for at least 10 min on each side.

### 2.3. Experimental Setup

Replicate Centre for Disease Control (CDC) bioreactors were used to develop biofilms on CS coupons over 2 weeks (Figure 1). Anaerobic conditions were generated by constant pure N_2_ gas injection. A solution temperature of 30 °C and gentle agitation were maintained using a stirring hotplate set to 50 rpm. Continuous nutrient replenishment was achieved using a 5 L reservoir cell connected to a peristaltic pump. The pump was calibrated to replace 30% of the reactor solution in each reactor every 24 h. ASW solution as described elsewhere [24,29] was used for the experiments with the following supplementation: 10 mM Bacto™ casamino acids (Thermo Fisher Scientific, Waltham, MA, USA), 10 mM sodium pyruvate, 10 mM D (+) glucose and 30 mM ammonium nitrate (NH_4_NO_3_). Nutrient concentrations were established to allow rapid progression of biofilm development without promoting transition of the population to the planktonic lifestyle. Sampling was conducted after 2 weeks of CS exposure to the consortium.

### 2.4. Control Assessment and Inhibitor Dosing

After 2 weeks of biofilm development, coupons were extracted for control CLSM, SEM, CFU and ATP measurements. Subsequently, clean, sterile bioreactors containing phosphate buffered saline (PBS, Sigma, St. Louis, MO, USA, pH 7.4) were used to apply treatments. In one reactor, PBS containing 1 mM of completely dissolved CTA-4OHcinn was applied. The remaining reactor contained the same solution with the addition of DNase-1 (100 ug/mL, Sigma). The reactor lids containing rods and remaining coupons were placed into fresh PBS (Sigma, pH 7.4) and then into the treatment reactors. Coupons were exposed to treatments for 4 h before transferring again into reactors containing fresh PBS (Sigma, pH 7.4). Finally, coupons were removed from rods for further processing and analysis.

### 2.5. Confocal Laser Scanning Microscopy (CLSM) and Post-Image Analysis

A Nikon A1+ confocal laser scanning microscope equipped with a 20 × dry objective lens was used to assess cell viability. The Filmtracer™ LIVE/DEAD™ Biofilm Viability Kit (propidium iodide and Syto9™, Thermo Fisher Scientific, Waltham, MA, USA) was purchased through Invitrogen™ (Thermo Fisher Scientific, Waltham, MA, USA) and mixed in ultrapure deionized water. The stain mixture was applied to coupons in 200 μL aliquots and incubated for 10 min before rinsing the coupon gently in PBS (Sigma, pH 7.4). Coupons were inverted and placed into a purpose-built dish with a central depression of radius 1 cm covered by a glass coverslip, to preserve biofilm architecture (ibidi^®^, Gräfelfing, Germany). Sequential micrographs were obtained using a 489.3 nm and 500–550 nm emission filter (Syto9™ imaging) and 561 nm laser with a 570–620 nm emission range (propidium iodide imaging). Separate tracks for emission and excitation paths were used in z-stack acquisition to minimize signal bleed-through. All experiments were conducted using the same microscope settings.

Micrographs were captured for visual representation of the control and treated samples, where the latest version of Nikon Elements software was applied for 3D reconstruction. Triplicate micrographs (z-stacks) were also captured from each surface and applied to calculate biovolume (μm^3^), % live biofilm and % dead biofilm composition using Bitplane (IMARIS), software (V9.8, Oxford Instruments plc, Tubney Woods, Abingdon, Oxon OX13 5QX, UK).

### 2.6. Scanning Electron Microscopy (SEM)

SEM was conducted using a Zeiss Neon field emission scanning electron microscope. Biofilm samples were rinsed and fixed as previously discussed [27]. Briefly, samples were rinsed gently in PBS (Sigma, pH 7.4) and incubated at 4 °C for 22 h in a 2.5% glutaraldehyde fixative solution. To minimize cell shrinkage, a temperature ramping stage was introduced to the fixation procedure. PBS and fixative solutions were prewarmed to 30 °C, and biofilms in pre-warmed fixative solution were allowed to gradually cool to room temperature before placing at 4 °C. Coupons were then removed from the fixative solution and dried under pure N_2_ gas for 12 h before sputter coating with 9 nm of platinum. An emission voltage of 5 kV was applied to the sample at a working distance starting at 4.5 mm. Microscopy was conducted using an in-lens secondary electron detector and an aperture size of 30 µm.

### 2.7. Standard Biofilm Colony Forming Unit (CFU) Enumeration

CFUs were extracted and quantified from control and treated coupons following existing standards [28]. To strip the biofilm from the coupon and homogenize the sample, coupons were placed into a tube containing 10 mL sterile PBS (Sigma, pH 7.4), vortexed for 30 s and sonicated for 10 s followed by 15 s on ice for a total of 2 min [27]. Agar plates containing the same nutrient composition as the reactor solution were produced using 15 g/L bacteriological agar. Plates were incubated at 30 °C for 2 days until visible colonies had appeared before counting.

### 2.8. Biofilm Adenosine Triphosphate (ATP) Assays

ATP is a nucleoside triphosphate involved in energy cycling for all life [30]. As the primary source of energy in bacterial cells, ATP is a prime biomarker for assessing biofilm viability. In the present research ATP was quantified from biofilms using the luciferase enzyme, which reacts with ATP to produce light correlating linearly with the ATP quantity in solution [30]. Total biofilm ATP was quantified using a Luminultra Quench-Gone Organic Modified ATP Test Kit and a Luminometer™ spectrophotometer (Hach, Loveland, CO, USA). Samples were prepared in tubes containing PBS (Sigma, pH 7.4) as described above. A standard calibration was performed before measurements were obtained.

## 3. Results

### 3.1. Biofilm Controls

Microscopic techniques were applied to assess the biofilm architecture and structure before treatment application. After two weeks of development, confocal laser scanning microscopy (CLSM) revealed a complex biofilm structure that was dominated by living cells (Figure 2A,C). A Bitplane (IMARIS) post-image analysis of control surfaces indicated a mean biovolume between 200 and 300 µM^3^ (Figure 3A). Between 65 and 75% of control biofilms were composed of living cells (green signal), with dead or damaged cells occupying 25–35% of the biovolume (Figure 3B).

Scanning electron microscopy (SEM) revealed the biofilm architecture, cell morphology and living arrangement in controls (Figure 4A,B). Micrographs captured a large surface area on control coupons (A) and indicated a complex, mature structure with dense cell arrangements and EPS. Through observation at higher resolution, various structures resembling cell morphologies were apparent (B), including single cells and long bacilli chains. Importantly, the dense living arrangement of cells within control biofilms was evident (Figure 4B). 

CLSM and SEM observations were supported by viability assays (Figure 5A), indicating that 1 × 10^8^ viable cells could be extracted per cm^2^ of coupon surface area in experimental (biological) replicates. Measurements of tATP are proportional to viability and were conducted to confirm CFU measurements (Figure 5B). In control experiments biofilm tATP extracted from coupons between 60,000 and 80,000 units/cm^2^, further confirming the viability of control biofilms.

### 3.2. Inhibitor Treatments

Treated biofilms demonstrated significantly different biofilm morphology and viability from controls. Both CTA-4OHcinn treated (B) and CTA-4OHcinn + DNase treated (D) biofilms had a reduced ‘green’ signal (corresponding to live cells). The ‘Red’ signal, corresponding to dead or damaged cells, was increased in all post-treatment CLSM.

Biofilm viability was not completely eliminated after 4 h of exposure to 1 mM CTA-4OHcinn alone (Figure 2B). When biofilms were exposed to both CTA-4OHcinn and DNase only the ‘red’ signal was detected in 3D reconstructions (Figure 2D), indicating enhanced degradation of biofilm architecture in the presence of an endonuclease.

A post-image analysis of CLSM micrographs indicated that after CTA-4OHcinn treatment, biofilm architecture remained relatively intact compared to controls (Figure 3A). However, viability was reduced by at least 40% (Figure 3B). In biofilms exposed to the dual treatment, biofilm structure was significantly reduced, as indicated by CLSM and post-image analysis (Figure 2D and Figure 3A). Although 3D reconstructions revealed little or no ‘green’ signal after CTA-4OHcinn + DNase treatments, viability remained in some areas as indicated by Figure 3B. A mean viability of 8–10% was detected, with dead or damaged cells representing 90% or more of the biofilm structure.

To support CLSM analysis, SEM was also conducted on coupons after 4 h of CTA-4OHcinn or dual CTA-4OHcinn and DNase treatments. Reduced biofilm density was observed in all treated samples compared to controls (Figure 4). Density of the biofilm living arrangement was also reduced (Figure 4D,F) compared to controls (Figure 4B). Lastly, cell arrangements were reduced from long chains to single bacilli. Although this observation could be the result of high stress, the authors recognize that these samples come from multi-species biological replicate experiments, where some morphological variation can be expected.

Finally, biofilm viability was assessed after treatment. All viability assays indicated a significant reduction in viability was induced by either treatment with CTA-4OHcinn or the dual CTA-4OHcinn and DNase treatment. In Figure 5A, a reduction in viability of two orders of magnitude (from 1 × 10^7^ to 1 × 10^5^) ca be observed after CTA-4OHcinn treatment. Microscopic results indicate that DNase assisted biofilm dispersal, allowing CTA-4OHcinn enhanced access, leading to greater efficacy. In CFU assays, a further log reduction in biofilm viability after exposure to the dual CTA-4OHcinn and DNase treatment was observed compared to the CTA-4OHcinn-only treatment, supporting microscopic analysis. Biofilm tATP (Figure 5B) demonstrated a similar trend, with a statistically significant reduction of viability compared to controls in both treatments. Finally, dual CTA-4OHcinn and DNase treated samples yielded less tATP than the CTA-4OHcinn-only treatment, which was statistically significant.

## 4. Discussion

Regardless of efficacy when first introduced into an engineered marine system, research indicates that multi-species biofilms develop chemical tolerance to almost any compound [25,31]. It is now critical to anticipate bacterial tolerance acquisition for enhanced long-term efficacy of biocidal compounds. By incorporating biofilm dispersal into dosing procedures, cells protected in EPS become directly exposed to chemical treatments. This approach has been successfully applied in the health industry [32] and is now considered for application in other industries affected by biofilm formation [31]. Guided by recent research implicating eDNA in marine multispecies biofilms developed on steel, it was hypothesized that eDNA degradation could enhance the function of a novel, multi-functional biocide compound.

Research involving clinically significant biofilms has so far pioneered the incorporation of dispersal mechanisms for viability reduction. Advancements in the understanding of clinically significant biofilms have developed rapidly; however, these communities are exposed to very different conditions compared to marine biofilms implicated in biofouling and MIC. In 2015, Okshevsky and colleagues identified eDNA as a target for biofilm disruption leading to enhancement of antibiotic efficacy [32], which was supported by a plethora of subsequent clinically-related studies [33,34,35]. Other promising dispersal targets to consider are emerging besides eDNA, such as extracellular proteins [36]. Research indicates that proteins have diverse roles within the biofilm, including facilitation of cell-cell cohesion and adhesion to surfaces [37], as well as cross-linking to eDNA structures [38]. While a combined enzymatic cocktail approach to dispersal seems logical based on the available literature, enzyme compatibility and target biofilm composition should also be considered. A recent study investigating antibiotic enhancement through EPS targeting revealed that tobramycin efficacy could be enhanced by application of either dispersin B (an enzyme that degrades 6-N-acetyl-d-glucosamine; PNAG) or DNase [35]. Interestingly a combination of these approaches was significantly less effective at enhancing the antibiotic treatment. Thus, co-treatment compromised enzyme activity. In the present study, a single enzymatic treatment was therefore selected to target mature biofilm dispersal. Since eDNA is an established ‘mortar’ in biofilms from other environments and was identified as a primary EPS component in previous research, biofilm eDNA was targeted in the present study using an endonuclease enzyme.

Biofilm composition and dispersal in native marine strains has been scarcely investigated in the scientific literature. In one 2021 publication, use of Peptide A (a 14-mer peptide structure resembling the Equinatoxin II protein) as a biofilm dispersal agent enhanced the efficacy of the biocide tetrakis hydroxymethyl phosphonium sulphate (THPS) [39]. Additionally, the aggressive MIC of a *Desulfovibrio* strain was reduced by up to 83% with Peptide A application [39]. The study supports biofilm dispersal mechanisms as viable options for targeted biofilm mitigation; however, the methods described do not consider the composition of the biofilm EPS.

In parallel with a biofilm dispersant, a novel, multifunctional biocide was applied at reduced concentrations to evaluate biofilm disruption. The multifunctional compound, CTA-4OHcinn is an effective corrosion inhibitor and biocide as demonstrated in previous reports [8,9]. CTA-4OHcinn was found to reduce bacterial viability during attachment stages by at least 99.7% at the recommended concentration (10 mM) [8]. Thus, the working concentration was reduced in the present research to 1 mM, and treatment time was optimized to preserve enough viability for treatment comparison.

Microscopic results confirmed that a complex biofilm had formed on CS after 2 weeks under the marine-simulating conditions evaluated. CLSM and post-image analysis further implied that biofilm parameters between experimental replicates were very similar (live and dead or damaged cells, biovolume; Figure 2 and Figure 4). SEM revealed a compact living arrangement and various cell morphologies, which is consistent with previous reports on multispecies communities [31,40]. Densely packed cell arrangements in the natural community promote beneficial behavior such as communication by quorum sensing, metabolic cooperation and horizontal gene transfer [41], which positively influences biofilm chemical tolerance. Compact, mature structures were also implied by CFU assays and tATP measurements. In the present research, these mature multi-species biofilm communities were considered baseline controls for biocidal treatment and treatment enhancement experiments.

After treatment with 1 mM CTA-4OHcinn, significantly reduced biofilm architecture and viability were observed. In CLSM results, a greater ‘red’ (dead) signal was recorded in z-stacks captured after treatment application (Figure 2), which was generally observed across replicates as indicated by post-image analysis (Figure 3). As previously expressed, a greater ‘red’ signal correlates with higher numbers of dead or membrane compromised cells [42]. SEM revealed a general reduction in biofilm material (Figure 4C) as well as a reduction in density (Figure 4D). This was expected, since CTA-4OHcinn contains the established antimicrobial hexadecyl trimethylammonium cation. The compound is expected to act on cell membranes to cause lysis based on research conducted on similar quaternary ammonium surfactants [43,44,45]. In CTA-4OHcinn, the quaternary ammonium cation is paired with the *trans*-4-hydroxy cinnamate anion to offer corrosion protection [9]. Thus, CTA-4OHcinn delivers at least one additional function along with reduced toxicity compared to CetBr.

Compared to controls and the CTA-4OHcinn-only treatment, the dual CTA-4OHcinn and DNase-1 treatment resulted in significantly reduced biofilm viability. Microscopic results showed a decrease in biofilm architecture and cell density (Figure 2 and Figure 4) which was supported by quantitative assays (Figure 5). Importantly, the results of dual treatments compared to controls indicate a significant structural role of eDNA in marine biofilms. As seen in previous studies in terrestrial environments, the present study demonstrates that eDNA can be considered a ‘building block’ of biofilms that can be targeted by endonucleases to disperse marine biofilms.

DNase-1 is an endonuclease that cleaves the phosphodiester backbone structure in double stranded DNA to produce smaller fragments [46]. Since DNase-1 is widely available, has a similar functional temperature range to bacterial growth conditions (30–40 °C), and shows promise as a dispersal agent in clinical biofilms, the enzyme was selected for dispersal of biofilms in the present investigation. DNase-1 enzymes are also able to continue functioning after reaction catalysis and are environmentally benign and easy to apply. Despite these benefits, limitations to large scale applications of enzymatic biocide enhancement still exist. Most importantly, mammalian endonucleases can be expensive to produce in large quantities. However, bacterial extracellular nucleases represent a practical solution to economic concerns [32]. Bacterial cells divide rapidly and can be cultivated affordably. Further, DNase-1 was applied in the present communication directly to ASW solution. Although the test solution contained critical Ca^2+^ and Mg^2+^ ions for DNase activity [46], bacteria also require these ions for growth and development processes [47]. Without the presence of both divalent ions, the activity of DNase-1 is known to be reduced [46]; thus, locations depleted of fresh seawater with high microbial activity pose a challenge to DNase-1 treatment. Here, we treated biofilms immersed in phosphate buffered saline (PBS), which was not supplemented with Ca^2+^ or Mg^2+^ ions, yet significant biofilm reductions were still observed with enzymatic treatment. Future investigations aim to further enhance enzymatic biofilm treatment efficacy through exploration of the optimal conditions for DNase-1 activity against biofilms in marine conditions.

The results of this study confirm that complex marine biofilms formed on CS after 2 weeks. Application of CTA-4OHcinn in 10 times reduced concentrations from the optimal recommended dose still demonstrated adverse effects on these biofilms, which were further enhanced by incorporation of an eDNA degrading stage. Incorporation of enzymes to assist with biofilm dispersal is a promising strategy for the effective and environmentally sensible control of pervasive marine biofilms.

## 5. Conclusions

To effectively manage marine biofilms with reduced environmental impact, a greater understanding of the EPS and community composition is required. The present research aimed to enhance the understanding of natural biofilm EPS by identifying the dominant structural matrix component under marine simulating conditions. Subsequently, through DNA and RNA-based sequencing analysis, this communication aimed to underpin changes to population dynamics over time and assess the origin of eDNA in marine biofilms. Microscopic analysis, post-image analysis and direct quantification of extracellular DNA (eDNA) suggest that eDNA is the most abundant and structurally important molecule in marine multispecies biofilms. Sequencing of eDNA, total biofilm DNA and RNA-based sequencing revealed that all originally inoculated bacterial strains contributed to the biofilm composition. The biofilm structure declined over time under limited, consistent metabolic supplementation, although cell viability and eDNA remained throughout the experiment. Interestingly the active fraction determined by RNA-based sequencing revealed that relatively stable communities form on carbon steel over 6 weeks of exposure. Lastly, no correlation was found between activity within the biofilm and DNA presence in the system.

## Figures and Tables

**Figure 1 microorganisms-10-01227-f001:**
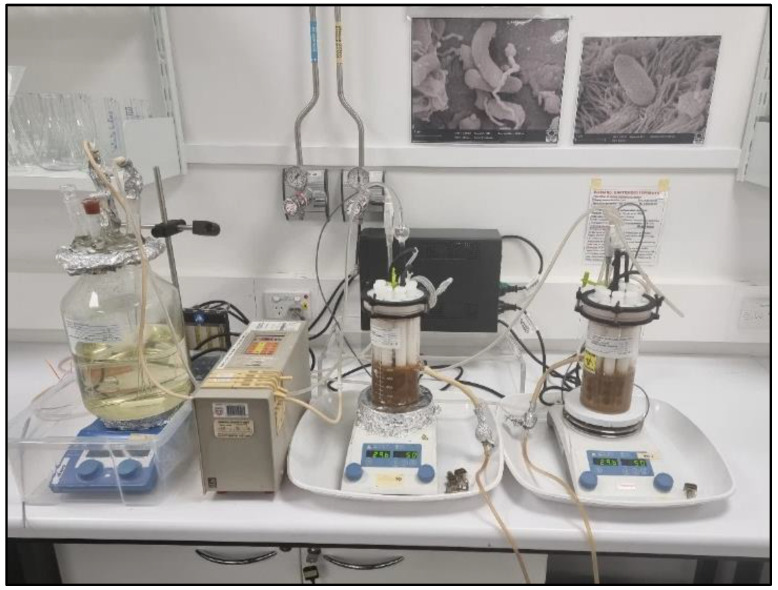
CDC reactor set-up with continuous flow for nutrient replenishment maintained using an external reservoir attached to a peristaltic pump, replacing 30% of the reactor solution every 24 h. Stirring hotplates were used to generate a constant temperature of 30 °C and agitation at 50 rpm. Reactors were constantly flushed with pure N_2_ gas to maintain anaerobic conditions.

**Figure 2 microorganisms-10-01227-f002:**
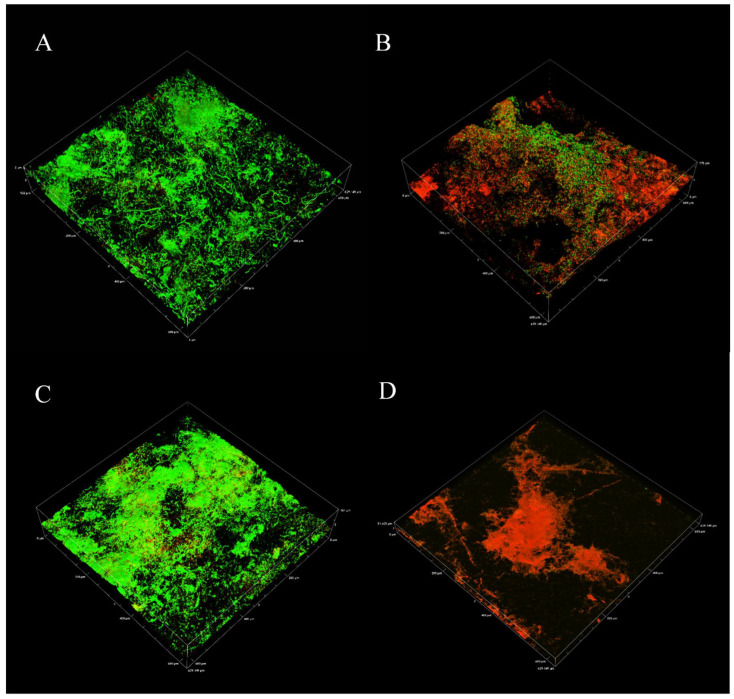
Confocal laser scanning micrographs showing live and dead cell in control biofilms compared to treatments, where: (**A**,**C**) = control biofilm, (**B**) = CTA-4OHcinn treated biofilm, and (**D**) = dual CTA-4OHcinn and DNase treated biofilm. Control micrographs (**A**,**C**) were captured from separate experiments before their respective treatments. Live cells are indicated by green, and dead or damaged cells are indicated by red.

**Figure 3 microorganisms-10-01227-f003:**
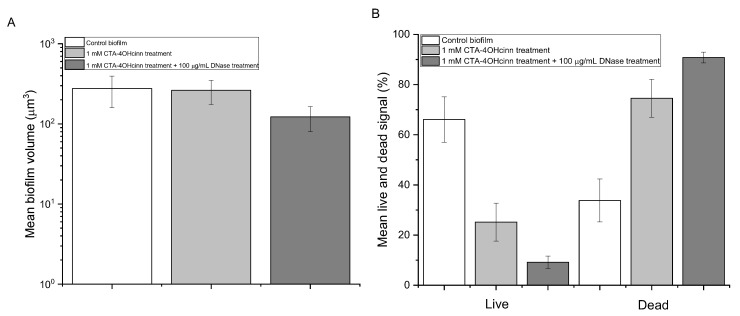
Mean biovolume as an average of five micrographs randomly captured across the coupon surface (**A**) and mean % live and dead cells from these micrographs (**B**) where the control biofilms (
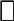
), are compared against CTA-4OHcinn treated biofilms (
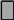
) and dual CTA-4OHcinn and DNase treated biofilms (
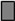
). Error bars represent the standard deviation of the data.

**Figure 4 microorganisms-10-01227-f004:**
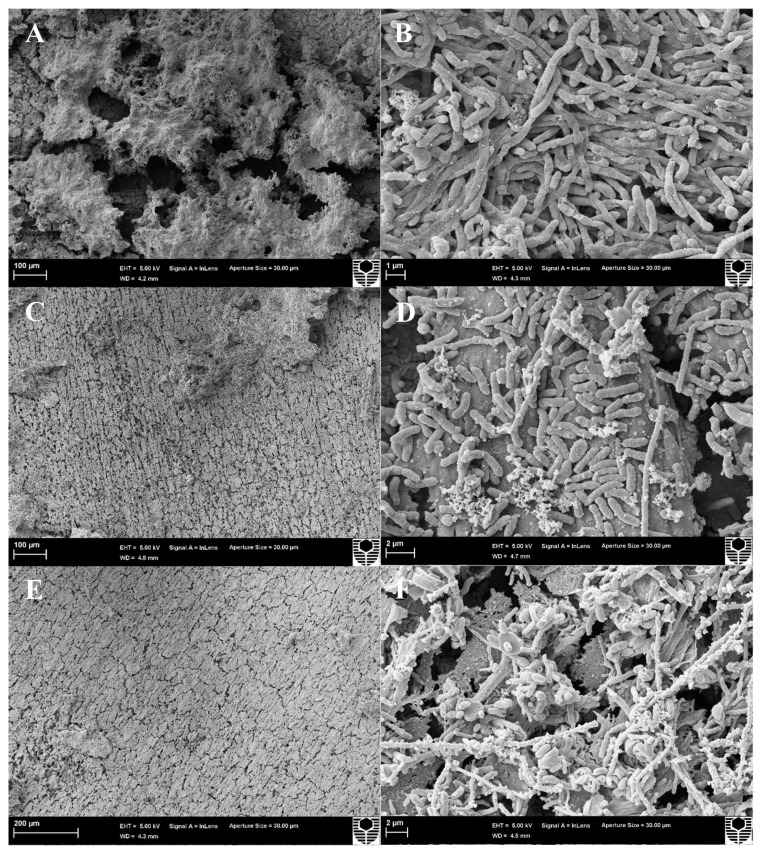
Scanning electron microscopy of biofilms before treatment (**A**,**B**), after treatment with 1 mM CTA-4OHcinn (**C**,**D**) and after dual treatment with 1 mM CTA-4OHcinn and DNase (**E**,**F**). Micrographs on the left of the figure depict the biofilm under low magnification to show a larger surface area, and micrographs on the left depict the cell arrangement within the biofilm.

**Figure 5 microorganisms-10-01227-f005:**
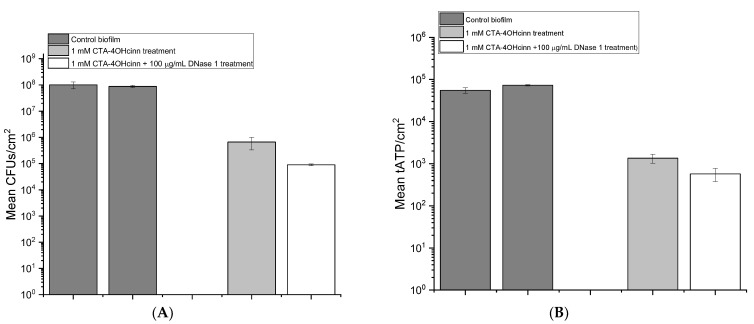
Biofilm viability assay colony forming units (CFUs; (**A**)) and total adenosine triphosphate (tATP; (**B**)) assays of control (
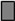
), 1 mM CTA-4OHcinn (
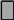
) and 1 mM 1 mM CTA-4OHcinn + DNase 1 enzyme (
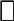
). Each bar represents measurements taken from triplicate biofilms with two technical replicates. Error bars represent the standard deviation of this data.

## Data Availability

Not applicable.

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
