# Peer review of "Enhancing Biocide Efficacy: Targeting Extracellular DNA for Marine Biofilm Disruption"

_microorganisms, 2022, doi:10.3390/microorganisms10061227_

Round 1

Reviewer 1 Report

I congratulate the authors on the work and manuscript. Biofouling is a major concern and current remediation methods are expensive and not very efficient. The manuscript is well written and the experimental design appropriate. The figures that are present (1-5) illustrate well the results, but the version I reviewed, unfortunately, does not contain Figures 6-10 which prevented me from fully appreciating the manuscript. The work can be of interest to readers of Microorganisms, especially in the way the authors used complementary approaches to study complex systems such as multispecies biofilms. The work also points to interesting combinations of organic compounds and enzymes to inhibit or reduce biofilm formation, a very promising field of investigation. I recommend acceptance for publication after a careful review of figures placement and their legend texts. Figure 4A and B can be improved by adding text or symbols for each column, and also a title for the access such as "treatments" or similar. Review the text of legends in Figures 2 and 4 for clarity. Make sure if symbols representing bar colors are used in parenthesis are used in the final version, since I could not see them in this PDF. lines 299 and 306: If AXP represents AMP, ADP and ATP, this can be detailed in the first mention of the term.

Author Response

Thank you for your comments, we appreciate your time reviewing our manuscript. 

Figure numbers were updated.

L220-222: description changed as suggested (in highlighted text)

L281-284: description changed as suggested (in highlighted text).

L286-390: discussion rewritten to connect with the remainder of the text and results. No Figure 6, rather we include the figures combined (1-5). 

Note: AXP data not relevant to this text. 

Thank you on belaf of all authors. 

Reviewer 2 Report

Comments:

Quite interesting article with the practical dimension. The analytical techniques used are adequate, although the problem requires more detailed information on qualitative and quantitative changes in biofilm (e.g. NGS analysis or Microfluidic Flow System Bioflux).

Extend the discussion of the your results to include scientific aspects, please.

Improve the graphic design of the article - enlarge the diagrams and the font on them. It's a pity photo # 1 is black and white. Correct the image numbers as well.

Author Response

Thank you for the time and effort taken to review the manuscript. Your comments were addressed as suggested, and here are our comments in no particular order: 

Figure numbers were inaccurate and have been corrected as suggested.

Figure 1 is no longer black and white (we found an original copy)

Figure legends have been updated (please refer to L220-222, and 281-284 in highlighted text). Colour bars are now included.

Discussion has been expanded and also significantly altered (please review). We believe this discussion fits better with the remainder of the article. 

Lastly, we agree that NGS would be beneficial to the project and manuscript, and has incorporated in our broader project. Of course, including sequencing analysis in the present manuscript was too much information, we believed, for the single document. In our previous research (submitted to this special issue) we utilise sequencing techniques, microscopic analysis and fluorimetric analysis to identify the connection between multispecies biofilms and EPS production, specifically in relation to extracellular DNA. We hope this fulfills a better picture for yourself and the broader readership. 

Best wishes on behalf of the authors. 
